# Marangoni-driven flower-like patterning of an evaporating drop spreading on a liquid substrate

F. Wodlei[1], J. Sebilleau[2], J. Magnaudet[2] & V. Pimienta[1]

Drop motility at liquid surfaces is attracting growing interest because of its potential applications in microfluidics and artificial cell design. Here we report the unique highly ordered pattern that sets in when a millimeter-size drop of dichloromethane spreads on an aqueous substrate under the influence of surface tension, both phases containing a surfactant. Evaporation induces a Marangoni flow that forces the development of a marked rim at the periphery of the spreading film. At some point this rim breaks up, giving rise to a ring of droplets, which modifies the aqueous phase properties in such a way that the film recoils. The process repeats itself, yielding regular large-amplitude pulsations. Wrinkles form at the film surface due to an evaporative instability. During the dewetting stage, they emit equally spaced radial strings of droplets which, combined with those previously expelled from the rim, make the top view of the system resemble a flower.

[1] Laboratoire des IMRCP, Université de Toulouse, CNRS UMR 5623, Université Paul Sabatier, 118 route de Narbonne, 31062 Toulouse, Cedex 9, France. [2] Institut de Mécanique des Fluides de Toulouse (IMFT), Université de Toulouse, CNRS, INPT, UPS, 31400 Toulouse, France. Correspondence and requests for materials should be addressed to V.P. (email: pimienta@chimie.ups-tlse.fr)

Nanometer- to millimeter-sized objects undergoing spontaneous directional motion, deformation, or topological changes, generically classified as active matter[1,2], are expected to find a broad range of applications in the miniaturization of many devices. In such systems, energy is supplied to the moving object by an external source[3] or by making it interact with its surroundings[4,5]. Drops spreading on a liquid substrate may fall into the second category when surface tension gradients take place, owing to phase change or to the presence of surfactants. Manifestations of the Marangoni stresses associated with these gradients are ubiquitous in liquid–liquid systems involving film spreading[6], dewetting[7], ligament break-up[8], interfacial bursting[9], or fingering[10]. These effects may even combine and give rise to highly organized patterns[11].

Under certain circumstances, the spreading of the drop may be followed by a dewetting stage[12–15]. The initial spreading results from the relative interfacial properties of the sub-phase and drop, the latter being either surface-active[13], or containing[14] or producing[15] (through a chemical reaction) a surfactant, which is then transferred to the supporting phase. This transfer modifies the surface properties in the drop surroundings, leading to its recoil. In most cases, these surroundings are definitively polluted and the recoiled state stays as the final permanent drop configuration. However, the expanding stage may sometimes repeat, provided that the drop environment has been regenerated, making it possible to restore the initial interfacial conditions. Few examples of such a behavior have been reported so far, the very first mention being for a cetyl alcohol drop on water[16]. Later, this observation was renewed with a surfactant-containing oil drop deposited on water[17]. Further studies of that system[18] revealed that the surfactant contaminates the water surface through local eruption events occurring at the drop edge, whereas regeneration results from surfactant evaporation at the aqueous surface. In binary systems, pulsations accompanied by the ejection of a ring of droplets were observed on a very short time scale for a cooled drop of 1-butanol deposited on a heated water phase[19]. Shape oscillations were recently reported with a drop of aniline deposited on an aqueous aniline solution[20]. Contamination of the drop surroundings was due to the ejection of aniline onto the free surface, the regeneration processes being attributed to periodical mixing of the aqueous phase ensured by Marangoni convection and initiated by the accumulation of aniline at the drop edge.

Here we consider a simple system made of a drop of dichloromethane (DCM) deposited on an aqueous substrate, with cetyltrimethylammonium bromide (CTAB), a common cationic surfactant[21], present in both phases with the same concentration. Interaction of the spreading drop with the supporting phase is dominated by DCM evaporation, which induces surface tension gradients, yielding the formation of a prominent rim and later a Marangoni instability. The drop and the film that surrounds it up to the rim exhibit the most regular large-amplitude pulsations ever reported. This film undergoes a very rich succession of hydrodynamic events (spreading, rim and wrinkles formation, rim break-up, and droplets ejection) with a degree of organization culminating during film recoil. Specific measurements and theoretical analysis enable us to establish the main characteristics of several of these steps and reveal the underlying driving physical mechanisms.

## Results

**Overall observations**. A 5.6 μl DCM drop is deposited onto a 25 ml aqueous phase filling a 7 cm Petri dish. CTAB is initially present in both phases with the same concentration (0.5 mmol l$^{-1}$). The drop life time is about 5–6 s but we focus attention on the dynamics observed typically during the first second. Although the whole system is chemically simple, its complexity arises from the physicochemical properties of the involved compounds: DCM undergoes evaporation and dissolution; it is heavier than water and surface active at the water/air interface; CTAB gives rise to adsorption/desorption processes at both water/air and water/oil interfaces.

As soon as the drop is released, the surface of the supporting aqueous solution deforms, generating capillary waves[22]. This initial wave train is immediately followed by a steep wave due to the spreading film arising from the drop. From that moment, the drop has the role of a reservoir with radius $R_D$ surrounded by a film with radius $R_F$ (Fig. 1a). Both $R_D$ and $R_F$ are found to oscillate over time, so that the whole system pulsates (Fig. 2). The generic pattern (see Methods and Supplementary Movie 1) observed during the first four pulsations (referred to as $P_1$–$P_4$ in Fig. 2) results in the occurrence of a highly complex pattern.

Although $R_D$ only slightly varies during $P_1$, expansion starts simultaneously at the drop and film edges during the following pulsations. However, the drop reaches its maximum radius before the film does, so that it recoils while the film is still growing. Expansion of the film is made visible by the rim that forms at its leading edge. The latter takes the form of a torus during $P_1$ (Fig. 1a), of a ring of closely packed beads during $P_2$ and $P_3$

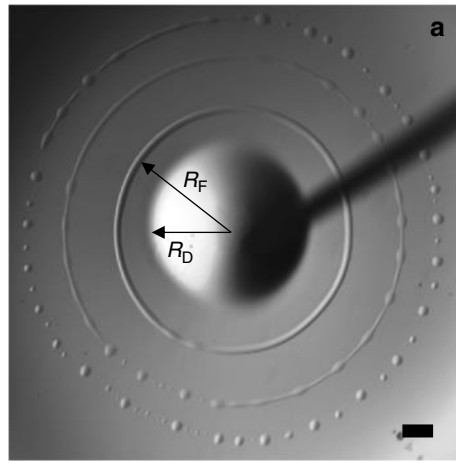
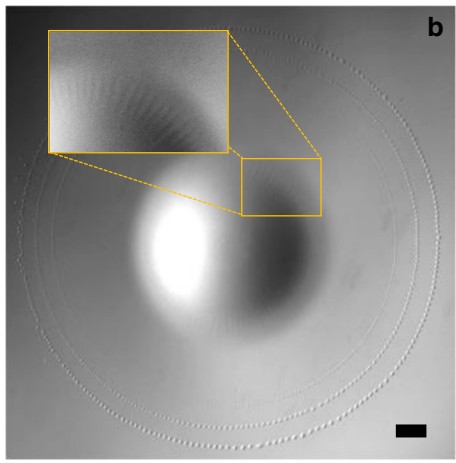

**Fig. 1** Expanding phase of a pulsation. **a** $P_1$, with three images shifted by 28.3 ms superimposed; arrows indicate the way $R_D$ and $R_F$ are defined in Fig. 2; **b** $P_3$, with three images shifted by 18.9 ms superimposed. The inset provides a zoom on the outer edge of the drop. Scale bars = 1 mm

(Fig. 1b), and eventually of a corrugated torus during $P_4$ (Fig. 3). Born at the edge of the drop during $P_1$, the rim occurs 3 mm apart from this edge during $P_2$, then 1.3 mm (resp. 0.5 mm) apart during $P_3$ (resp. $P_4$). Based on the evolution of this initial distance, the sequence should rather be ordered as $P_2$–$P_3$–$P_4$–$P_1$.

In a later stage, the rim breaks up into a large number of droplets. The number of ejected droplets yields the same ordering (with 560, 240, 120, and 30 ejecta for $P_2$, $P_3$, $P_4$, and $P_1$, respectively) and so does their diameter (75, 100, 240, and 340 μm, respectively). The size dispersity of ejected droplets also evolves according to the same sequence: the droplet ring is perfectly monodisperse during $P_2$, with a tiny satellite in between two droplets during $P_3$, with an increasing number of satellites during $P_4$ and $P_1$. The thicker the expanding rim, the larger the number of satellites[23].

The velocity of the advancing rim slowly decreases over time. Its typical magnitude is ~45 mm s$^{-1}$ during pulsation $P_1$ and 30 mm s$^{-1}$ during the next three pulsations. Variations in $R_D$ stay below measurement accuracy during $P_1$, increase during $P_2$ and $P_3$, and become significant during $P_4$. There the drop grows for a while with a radial velocity about 10 mm s$^{-1}$. Its recoil is milder, with a velocity approximately three times smaller. The drop shrinks until it reaches a minimum radius which then remains constant for a while, in relation with the dewetting dynamics of the surrounding film. Radial stripes develop in the peripheral region of the drop surface during the spreading stage of $P_3$

(Fig. 1b) and $P_4$ (Fig. 3). They do not occur during previous pulsations (compare with Fig. 1a). Although they remain confined within the drop during $P_3$, they become much more prominent during $P_4$ (Fig. 3), developing over the surface of the expanding film that now exhibits regularly-spaced radial wrinkles. As will be shown below, these corrugations have a key role in the generation of the highly ordered pattern that sets in while the film recoils during $P_4$ and following pulsations.

Very shortly after the rim breaks up, a new ring of droplets forms at the leading edge of the film left behind it (Fig. 4). These droplets have a much smaller size than those previously emitted by the advancing rim. During the first three pulsations, the two rings coexist for some time before the tiny droplets that form the inner ring disappear through dissolution and evaporation.

By that time, the film has started to recede. It then becomes hardly visible during $P_1$, $P_2$, and $P_3$, when it merely appears as a fuzzy shadow; its leading edge deforms while shrinking and fades away. The dewetting pattern observed during $P_4$ differs dramatically, owing to the presence of the radial wrinkles. Similar to what happens during previous pulsations, a second ring of droplets forms at the leading edge just after the advancing rim has broken up. However, the size of the corresponding droplets is now non-uniform, bigger droplets being emitted at positions corresponding to wrinkle tips. Although the tiny droplets generated in between two wrinkles quickly dissolve, the bigger ones emitted at wrinkle tips survive for a much longer time. Then

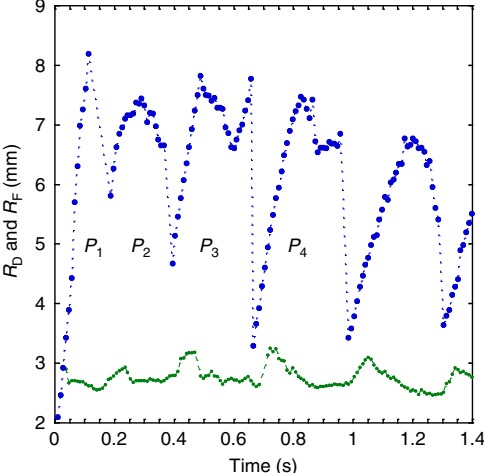

**Fig. 2** Film and central reservoir radius variations vs. time. $R_D$ (green curve) and $R_F$ (blue curve) during pulsations $P_1$–$P_4$; dotted line: guide-to-the-eye between experimental points

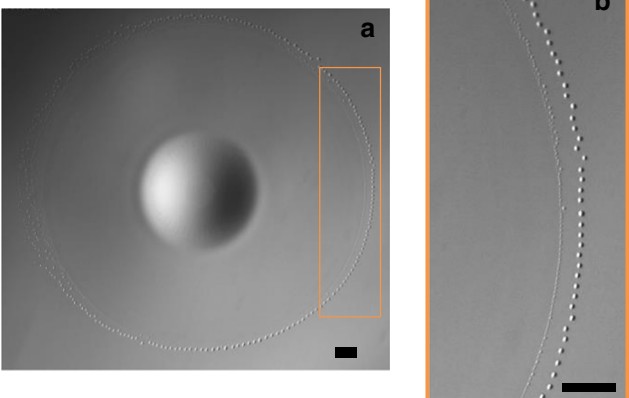

**Fig. 4** Two rings of droplets coexist for some time just before the film recoils during pulsation $P_3$. The inner ring, with droplets ~35 μm in diameter, occurs shortly after the expanding rim broke up. **a** General view; **b** magnification of the framed area; scale bars = 1 mm

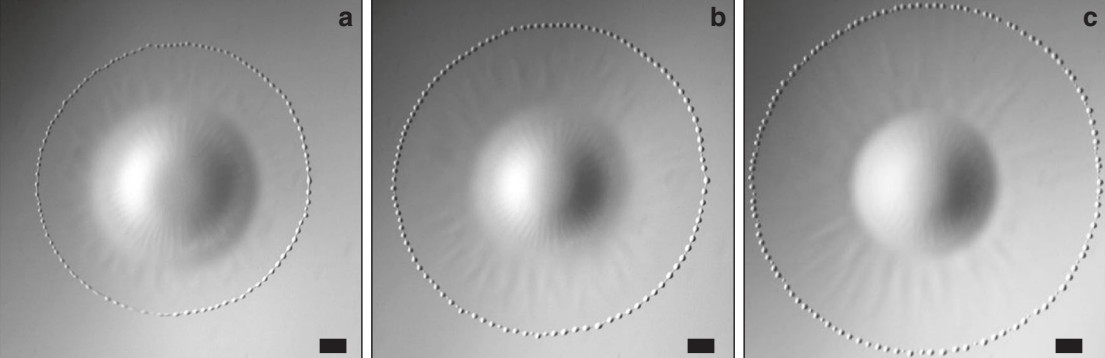

**Fig. 3** Expanding stage of pulsation $P_4$. **a** at time $t$; **b** at time $t + 28.3$ ms; **c** at time $t + 56.6$ ms. Scale bar = 1 mm

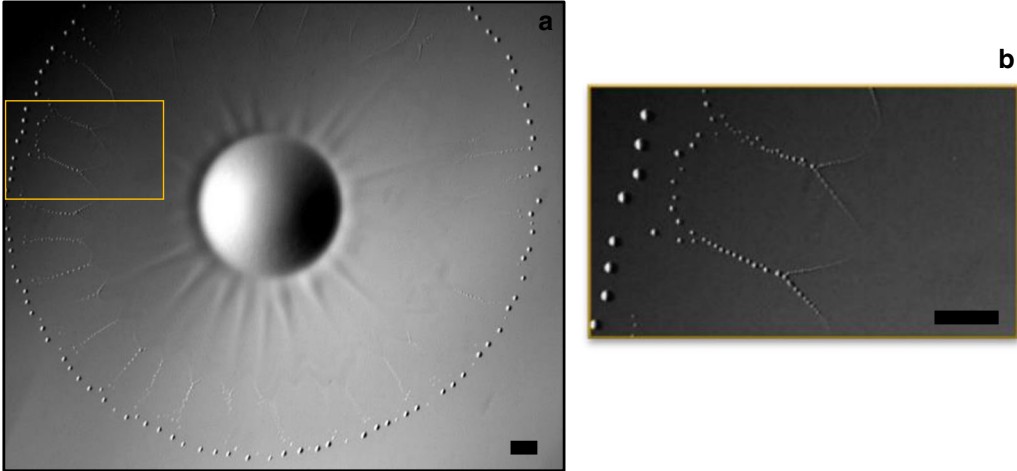

**Fig. 5** Dewetting flower pattern observed during film retraction of pulsation $P_4$. **a** General view; **b** magnification of the framed area; scale bars = 1 mm

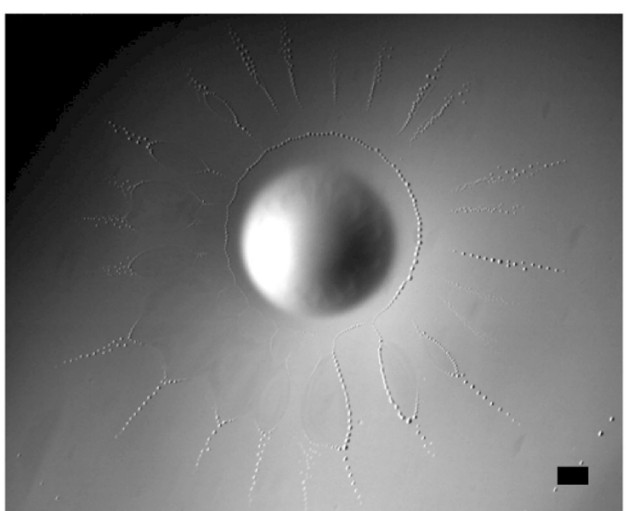

**Fig. 6** Islands formed during the late dewetting stage of $P_4$. These islands coexist with the advancing rim formed at the beginning of $P_5$; scale bar = 1 mm

two series of droplets, one on each side of a wrinkle tip, are generated. At that point, the leading edge of the recoiling film exhibits a zigzagging shape made of tips and troughs connected by strings of droplets. As the troughs go on recoiling, these strings keep on lengthening. Droplets generated on each side of a tip continuously slide toward it, where they merge (Fig. 5a,b).

Droplets are emitted from the tips in a very ordered way (Fig. 5b and Supplementary Movie 2). When three to four of them have merged at a tip, the resulting daughter drop (about 90 μm in diameter) is ejected outward. While tips recoil, troughs widen more and more. At some point, two consecutive troughs merge at the back of a wrinkle, isolating the corresponding tip and strings of droplets from the reservoir drop. Then, when the next pulsation takes place, a new rim forms in the neighborhood of the central reservoir, whereas 'islands' encircled by tiny droplets and extended outward by a long spike of bigger droplets subsist further away before disappearing (Fig. 6).

**Spreading dynamics**. The spreading coefficient, $S_\infty = \gamma_{water/air} - (\gamma_{oil/air} + \gamma_{oil/water})$, built on the three interfacial tensions involved in the system, drives its initial behavior[24]. Here (see Methods),

the respective values of the equilibrium surface tensions lead to a positive $S_\infty$, in agreement with the observed initial expansion. Figure 7a, b show how the film radius, $R_F(t)$, evolves during the expanding stage of pulsations $P_1$ and $P_4$, respectively (the behavior is similar to that of $P_4$ during $P_2$ and $P_3$). During $P_1$, the spreading dynamics is essentially characterized by a single exponent, $n \approx 1.10$. In contrast, two distinct stages are observed during subsequent pulsations. The spreading is significantly faster during the first of them, with for $P_4$ an exponent close to 1.20, then decreasing to 0.82 during the second half of the expansion.

The latter exponent is close to the classical theoretical prediction $n = 3/4$ corresponding to the surface-tension-driven spreading of a thin film in the case where the net driving force is balanced by the viscous force that develops in the boundary layer of the supporting fluid[25–27]. The larger exponent noticed during $P_1$ stems from the reorganization of the CTAB distribution at the water surface just after the drop has been released. This reorganization takes place because the minimum area per adsorbed molecule is 80 Å$^2$ at the DCM/water interface, whereas it is only 53 Å$^2$ at the air/water interface[28]. Hence, when the drop is released, CTAB molecules initially present on the part of the water surface that comes in contact with the drop are swept away, as sketched in Fig. 8. They accumulate provisionally ahead of the edge of the spreading film, until they are redistributed uniformly at the water surface.

This sudden accumulation of CTAB molecules makes the local value of the water/air surface tension fall temporarily below the equilibrium value corresponding to the nominal CTAB concentration of 0.5 mmol l$^{-1}$. Because of this, the spreading coefficient also falls from its equilibrium value, $S_\infty$, to a smaller initial value, $S(t = 0) = S_\infty - \Delta S$ with $\Delta S > 0$. Relaxation of $S(t)$ toward $S_\infty$ is dominated by desorption within the aqueous phase (surface diffusion corresponding to a much longer characteristic time and convective motions in water being weak). As shown in the Methods section, desorption makes the spreading coefficient relax exponentially toward its equilibrium value at a rate determined by the effective desorption coefficient, $k_{eff}$. With a bulk concentration of 0.5 mmol l$^{-1}$, $k_{eff}$ stands in the range 0.5–1 s$^{-1}$ for CTAB[29], so that the exponential relaxation is quasi-linear throughout $P_1$ with $S(t) \approx S_\infty - \Delta S + (k_{eff}\Delta S)t$. Balancing the net surface tension force with the viscous resisting force then implies that the evolution of the film radius obeys (see Methods).

$$R_F(t) \propto (\rho_w \mu_w)^{-1/4}\{(S_\infty - \Delta S)^{1/2} t^{3/4} + (k_{eff}\Delta S)^{1/2} t^{5/4}\} \quad (1)$$

where $\rho_w$ and $\mu_w$ stand for the water density and viscosity,

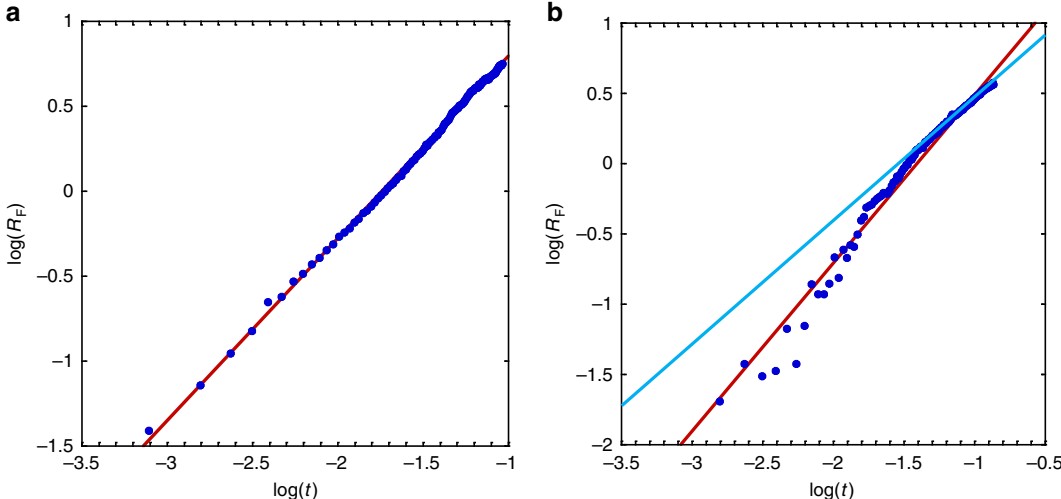

**Fig. 7** Position of the leading edge of the film during the expansion stage as a function of time. **a** Pulsation $P_1$; **b** pulsation $P_4$. In **a**, the slope of the red line is 1.1. In **b**, the slopes of the red and blue lines are 1.2 and 0.82, respectively

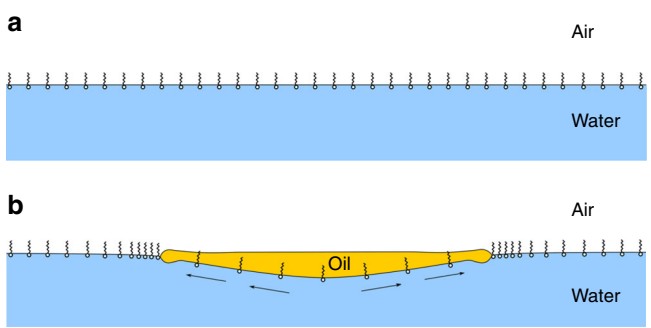

**Fig. 8** Sketch of the early evolution of the CTAB distribution resulting from the deposition of the drop. **a** The surface concentration of CTAB at the air/water interface is initially uniform; **b** the minimum area per adsorbed molecule being larger at the oil/water interface, CTAB is swept away and accumulates provisionally ahead of the film, forcing the surface tension $\gamma_{water/air}$ to decrease abruptly around the contact line

respectively. In agreement with the observed exponent $n = 1.1$, the 'super-spreading' law in equation (1) indicates that the film spreads faster than predicted by the classical $t^{3/4}$ law (recovered when $\Delta S \rightarrow 0$) during $P_1$. The exact exponent of the corresponding law stands in the range 3/4–5/4 and is determined by $k_{eff}$ and by the (unknown) relative initial drop of the spreading coefficient, $\Delta S/S_\infty$. During the recoil stage of subsequent pulsations, another physico-chemical reorganization process (to be described later) takes place within the aqueous phase. Its outcome is that, starting from the negative value characterizing the recoil stage, the spreading coefficient gradually recovers its positive equilibrium value, $S_\infty$. Spreading starts as soon as $S(t)$ reverses to a positive value, which corresponds to $\Delta S = S_\infty$ in equation (1). For this reason, only the last term in equation (1) is expected to exist in that case, making the spreading exponent close to 5/4. This is what the behaviour observed in Fig. 7b for $P_4$ confirms: $n$ is then closer to the theoretical prediction $n = 5/4$ than during $P_1$ (Fig. 7a), whereas, during the second stage when $S(t)$ gets close to its equilibrium value $S_\infty$, it approaches the theoretical value $n = 3/4$ expected for a constant spreading coefficient.

**Evaporation and rim formation**. The advancing rim at the leading edge of the film looks similar to the Marangoni ridge observed when a film surface is covered with a surfactant[22,30–32] or when a thin volatile film evaporates at the surface of a deep fluid layer[33]. The present system falls in the second category, DCM evaporation being responsible for the rim formation. Indeed, we performed complementary experiments with a DCM drop of same volume (5.6 µl) without CTAB inside and observed a similar rim, which allows us to rule out the role of the surfactant. Focusing on evaporation, we determined the averaged evaporation rate per surface unit, $Q_{ev}$, by depositing DCM drops of various volumes and recording the evolution of the total mass of liquid enclosed in containers with various air/DCM contact areas. This procedure (see Methods), revealed that this mass decreases linearly over most of the drop life time at a surface rate $Q_{ev} \approx 2 \times 10^{-4}$ mg mm$^{-2}$ s$^{-1}$, corresponding to a surface heat flux $J_0 \approx 70$ W m$^{-2}$. These findings may be used to obtain some insight into the dynamical properties of the system, provided that the temperature and velocity distributions within the film are known. Approximate solutions were derived for both of them (see Methods). This derivation shows that a negative radial temperature gradient increasing linearly with the distance to the drop takes place within the film. This gradient results in a linear increase of the oil/air surface tension from the edge of the drop to that of the film, which generates an outward Marangoni flow in the latter. With the surface temperature distribution at hand, we determined the velocity field corresponding to this flow (see Methods). Beyond the leading edge of the macroscopic spreading film considered up to now, a precursor DCM film stands at the water surface. Evaporation also takes place at the surface of that film, the area of which is much larger than that of the macroscopic spreading film (see Methods). However, convective motions are weak in the aqueous phase, so that this film is almost stagnant and cools progressively to supply the energy consumed by evaporation. In contrast, within the macroscopic spreading film, energy is supplied by the drop and transported outward by the spreading flow. At the contact line, the non-zero shear stress corresponding to the surface tension gradient at the surface of the spreading film has to match with the shear-free condition prevailing at the air/precursor film surface. This abrupt change is accommodated by the rim, the large curvature of which locally induces a radial velocity correction that cancels the Marangoni

contribution[34]. This condition determines the rim thickness which may be measured from top views such as those of Fig. 1. As shown in the Methods section, combining this thickness with the above condition provides an indirect manner of estimating the thickness of the spreading film at the inner edge of the rim. Considering the first of the three spreading stages displayed in Fig. 1a, b, this approach yields values of the local film thickness close to 3 μm and 1 μm for $P_1$ and $P_4$, respectively.

**Rim break-up**. The topological transition by which the rim turns into a ring of droplets during the expansion stage of each pulsation is reminiscent of the Rayleigh–Plateau capillary instability experienced by cylindrical liquid threads and jets. In the usual case of a stationary cylindrical thread with a constant undisturbed radius $a$, the linear stability theory[35] predicts that, provided the influence of viscous effects within the thread and those of inertia and viscosity in the surrounding fluid are negligible, the most amplified wavelength, $\lambda_s$, is such that $\lambda_s \approx 9.01a$. However, in the situation considered here, the capillary instability acts on a rim that expands continuously, which implies a local stretching of the fluid. Hence, the above prediction does not apply directly. In particular, the most amplified wavelength, $\lambda$, is not constant but instead increases over time. This may be observed by eye in Fig. 1a by comparing the wavelengths of the varicosities that are well visible at the first two rim positions.

In the Methods section, we show that the evolution of the disturbances that grow at the rim surface comprises two distinct stages. During the first of them, several mechanisms act to select the disturbance that eventually becomes most amplified. Once this disturbance has emerged, its characteristics evolve in such a way that the number of wavelengths along the rim stays constant, as requested by the $2\pi$ periodicity of the circular geometry. During that second stage, $\lambda(t)$ grows proportionally to $R_F(t)$. The first rim position displayed in Fig. 1a corresponds to the end of the first stage, making it the proper configuration to be compared with the relevant theory[36, 37]. This comparison, which involves the characteristic times associated with capillary, viscous, and stretching effects, plus the origin of time at which the rim gets its final volume, yields the results summarized in the Supplementary Table 1. A remarkable agreement between the predicted and measured values of $\lambda/a$ is found, providing a strong argument to interpret the rim break-up as resulting from the Rayleigh–Plateau mechanism in the presence of weak but non-negligible stretching and viscous effects.

**Spreading–recoil and recoil–spreading transition mechanisms**. Up to this point, one may expect the final state of the system to correspond to complete wetting. However, in surfactant-containing systems, the initial expanding phase may sometimes be followed by retraction[13,14,15], because the deposited drop modifies its own neighbourhood, which in turn changes one or possibly two of the surface tensions involved in the system, making the initially positive $S$ switch to a negative value. As $\gamma_{oil/air}$ is not expected to vary significantly in the present case (besides its slight radial increase due to the negative radial temperature gradient), a negative $S$ can be reached only if $\gamma_{water/air}$ decreases or/and $\gamma_{oil/water}$ increases. Indeed, DCM spread on the water surface by droplets resulting from the rim break-up significantly decreases $\gamma_{water/air}$ (see Supplementary Fig. 1). Moreover, the DCM/water interfacial area increasing by a factor of typically six during the spreading stage, the CTAB interfacial concentration decreases by the same amount, leading to an increase of $\gamma_{oil/water}$[28]. These two cooperative effects represent the main driving causes of the recoil process.

For such a sequence to repeat and give rise to successive pulsations, recovery processes are compulsory to bring the system back to its initial state. Film recoil restores the initial value of $\gamma_{oil/water}$, owing to the reduction of the corresponding interfacial area. DCM evaporation at the surface of the recoiling film contributes to make $\gamma_{water/air}$ re-increase. Last, the present ternary system is known to be prone to form an oil-in-water microemulsion in which surfactant-surrounded buds of oil detach from the interface[28]. This process ease the transfer of DCM into the aqueous phase, also contributing to restore the initial value of $\gamma_{water/air}$[38].

Hence, after each surface contamination by the CTAB-containing DCM film, these three mechanisms altogether allow the system to gradually meet conditions making a new pulsation able to start. Although qualitative, this description provides the overall explanation for the observed pulsating behaviour which, compared to the scarce examples present in the literature[19,39,40], appears to be unique, given the amplitude, regularity, and symmetry of the repeated pulsations.

**Recoil during the first three pulsations**. In each pulsation, after the advancing rim has broken up, $S(t)$ decreases before turning negative. During this transient, the temperature distribution within the film is still close to that reached at the end of the spreading stage, so that a negative radial temperature gradient still exists at its surface. Hence, an outward Marangoni flow subsists and must still match at the new leading edge with the shear-free condition prevailing at the surface of the aqueous phase. This matching requires the formation of a new rim. However, the radius of that rim is much less than that of the primary one, as most of the DCM supplied by the reservoir drop during the spreading stage has gone with the droplets emitted by the primary rim. As the smaller the rim radius the shorter the break-up time, this new rim breaks up within an extremely short time, making it virtually invisible as such. In contrast, as shown in Fig. 4, its consequences are well visible in the form of the inner ring of droplets resulting from its break-up. Based on the size of these droplets and on known characteristics of the Rayleigh–Plateau instability, an estimate of the rim break-up time may be obtained. It predicts that break-up takes place within 0.4 and 0.1 ms during $P_1$ and $P_3$, respectively, which confirms the above assumption (see Methods).

During these first three pulsations, the two rings coexist for some time before the tiny droplets emitted by the second of them dissolve in the aqueous phase. The process just described could in principle repeat itself. However, the film, which has started recoiling in the meantime, is becoming very thin after the second ring has been emitted. Moreover, the recoil flow acts to lower the radial temperature gradient established during the spreading period, weakening the Marangoni effect. This is why no new discernible rim actually forms during the recoil stage.

**Wrinkles formation**. The radial stripes observed at the edge of the drop during the expanding phase of $P_3$ and $P_4$ are reminiscent of the longitudinal rolls that develop in thin thermocapillarity driven films[41,42]. However, during $P_3$, these stripes do not seem to have any impact on the film, the surface of which stays smooth. In contrast, prominent wrinkles stand at the surface of the expanding film during $P_4$. Nevertheless, their connection with the stripes present on the peripheral part of the drop surface is unclear, as the number of the latter is typically twice as large as that of the wrinkles. These wrinkles look similar to those observed at the surface of evaporating films climbing on an inclined plane[43,44]. In the latter reference, it was shown that the

underlying evaporative instability is driven by the competition between the Marangoni stress and the restoring gravity effect. However, besides the consequences induced by the two different geometries (circular vs. planar), two main differences exist between the present configuration and the one considered in that reference. First, the film motion is driven by the antagonistic effects of thermocapillarity and gravity in the latter, whereas it is almost controlled by spreading in our system, the Marangoni flow only providing a small correction. Second, the evaporation flux results from compositional gradients rather than temperature gradients in ref. [44], as in most experiments carried out with miscible fluid setups. Thermal diffusivity being typically two orders of magnitude larger than molecular diffusivity in liquid systems, this difference has a deep impact on the conditions under which wrinkles may develop.

In the Methods section, we adapt the approach developed in ref. [44] to present conditions. The corresponding linear stability analysis shows that wrinkles can grow as soon as the film thickness at the edge of the drop stands below a critical value, $h_c$, which we predict to be ~5–6 μm. This is slightly more than the film thickness estimated for $P_1$ ($\approx 3$ μm), so that wrinkles could in principle be present during that pulsation. However, the stability analysis also predicts that the most amplified wavenumber (hence the approximate number of wrinkles, $N_W$) that can be accommodated along the drop perimeter, varies as $(h_c/h_1 - 1)^{1/2}$, where $h_1$ stands for the undisturbed film thickness at the edge of the drop. Thus, $N_W$ slowly increases with $h_c/h_1$ and the predicted wavelength during $P_1$ is still larger than the drop perimeter, in line with the absence of wrinkles in Fig. 1a. Predictions suggest that $N_W$ stands between 1 and 2 during $P_3$. With such large wavelengths, the deflections of the surface are too small to be discernible on top views such as that of Fig. 1b, so that it may well be that one or two wrinkles are present but simply cannot be detected on the images. Last, the film being much thinner during $P_4$ ($h_1 \approx 0.2$ μm), the stability analysis then predicts $N_W = 7$, which corresponds to an angle close to 50° per wavelength. Although significant, this number of wrinkles is three to four times less than what is seen in Fig. 5, where ~ 25 of them can be identified. There may be numerous explanations to this underestimate, given the various assumptions involved in the model, especially in the estimate of the film thickness, and the significant uncertainty on the experimental value of the local evaporation flux. The observed pattern may also simply be out of reach of linear theory, as the height of the wrinkles displayed in Fig. 5 is presumably of the same order if not larger than the undisturbed film thickness. Despite the quantitative difference in $N_W$ during $P_4$, the above comparison supports the overall scenario that wrinkles result from a purely thermally driven evaporative instability, which, compared with its more common solutal counterpart[43,44], requires much thinner films to set in. The predicted critical film thickness and increasing number of wrinkles as the film thins down are consistent with the behaviors observed during the successive pulsations.

## Discussion

Wrinkles have a crucial role throughout the recoil stage of $P_4$. The beginning of the sequence is similar to that observed during previous pulsations, with the generation of a second, much thinner rim almost immediately disintegrating in an inner ring of droplets. The presence of the wrinkles first manifests itself in the non-uniform size of these droplets. As the recoil starts, their effects become more spectacular with the occurrence of the zig–zag pattern observed at the edge of the

receding film (Fig. 5). Indeed, although evaporation takes place everywhere at the film surface at an approximately uniform rate, the thinner the film the shorter its takes to evaporate it completely, which makes the dewetting front propagate faster in between wrinkles. Then, each side of a wrinkle, from the tip to the trough, becomes a new contact line between the DCM film and the aqueous phase. Clearly, this contact line is almost perpendicular to the recoil motion in the tip region. Moreover, inspection of the cellular flow within a wrinkle (equations (11)–(13) in the Methods section) indicates that, at the DCM/air surface, this flow structure brings DCM laterally from wrinkle crests to that contact line. Thus, in addition to its dominant inward radial component, the velocity field within a wrinkle also comprises a circumferential component almost perpendicular to the wrinkle's sides. Hence, DCM now moves with a velocity having a component perpendicular to the contact line all along the wrinkle periphery. Once again, a rim must set in to accommodate this situation and, being very thin, it becomes almost immediately unstable and degenerates into droplets. Bigger droplets being emitted at the tip due to the merging and coalescence process already described, their outward motion drags smaller droplets generated along each side of the wrinkle toward its tip. This process shortens and thins the wrinkle, so that a new contact line emerges, forcing a new rim to occur. The process repeats itself as far as the temperature distribution sustains a significant Marangoni flow within the wrinkles.

The individual physical and chemical mechanisms dissected in the previous section draw a consistent picture of the phenomena that combine in the present system to produce its complex and highly organized behavior culminating in the flower-like pattern displayed in Fig. 5. We were able to establish most of them, qualitatively or sometimes quantitatively, through the dedicated experiments and models described in the Methods section. However, some of these mechanisms still have to be more firmly established, especially those involving the wrinkles during the recoil stage. This requires a specific study, presumably with the help of numerical simulations of the fully coupled equations governing mass, momentum, and energy conservation. Although the role of CTAB appears instrumental in helping the system to come back to its initial state after each pulsation, the fascinating highly ordered pattern revealed by present experiments results mostly from DCM evaporation, which, combined with surface-tension-driven spreading, generates a Marangoni flow that gives rise to the formation of a prominent rim and later to the occurrence of wrinkles through a purely thermo-hydrodynamical instability mechanism.

## Methods

**Reagents and physico-chemical properties**. All chemical reagents used are of analytical grade: CTAB (Aldrich, ≥ 99%) and DCM (Aldrich, High Pressure Liquid Chromatography grade). Water is ultra-pure (resistivity > 17 MΩ cm). All experiments are carried out at room temperature. The CTAB concentration is 0.5 mmol l$^{-1}$, both in the aqueous phase and in the drop.

The density of DCM (1.33 g ml$^{-1}$) is larger than that of water. The solubility of water in DCM is 2 g l$^{-1}$ at 25 °C. The solubility of DCM in pure water is 13 g l$^{-1}$, which corresponds to a volume of 250 μl in 25 ml. The solubility of CTAB in water is 15 g l$^{-1}$ (41 mmol l$^{-1}$). The boiling point of DCM is 39.6 °C.

Determination of equilibrium values of interfacial tensions with the pendant drop technique provides $\gamma_{water/air} = 47$ mN m$^{-1}$ in a 0.5 mmol l$^{-1}$-CTAB aqueous solution, $\gamma_{oil/air} = 28$ mN m$^{-1}$ at the DCM/air interface, and $\gamma_{oil/water} = 2.5$ mN m$^{-1}$ at partition equilibrium (0.5 mmol l$^{-1}$ CTAB in the aqueous phase).

The spreading coefficient is then $S = 47 − (28 + 2.5) = 16.5$ mN m$^{-1} > 0$. This value is certainly overestimated, as the existence of the precursor film of DCM that extends ahead of the rim at the water surface tends to decrease $\gamma_{water/air}$. Moreover, due to DCM dissolution and film expansion, the effective value of $\gamma_{oil/water}$ under

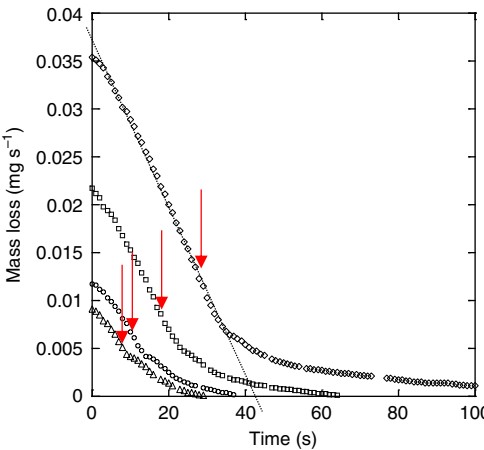

**Fig. 9** Mass loss during the evaporation of DCM drops with 0.5 mmol l$^{-1}$ CTAB in a 7 cm Petri dish. Initial drop volumes from bottom to top are 6, 9, 16.6, and 26.5 µl, respectively. Arrows indicate the time at which the drop disappears. The dotted line is used to determine the evaporation rate

present non-equilibrium conditions is expected to be higher than the above equilibrium value. However, in any case, $S$ keeps a positive value.

**Processing techniques**. Visualization is obtained using an optical Schlieren set-up. Images are recorded with a high-speed PCO Dimax camera at full resolution (2,016 × 2,016 pixels) with an acquisition rate of 1,279.35 fps and a field of view of 22 mm × 22 mm. Image processing is performed using the ImageJ open source software.

**Measurements and reproducibility**. The CTAB solution (25 ml) is poured into a cylindrical container (diameter 70 mm). Then a single drop of DCM is carefully deposited onto the surface of the solution using a gastight syringe. The drop volume is 5.6 ± 0.1 µl. We performed 19 experiments under similar conditions; 14 of them exhibited more than 4 pulsations. The pattern observed during the first four pulsations ($P_1$–$P_4$) led to the highly ordered dewetting structure described above (during $P_4$) 18 times out of 19; this structure qualitatively survived during the following pulsations if any.

**Determination of the evaporation rate**. DCM evaporation was measured by placing the system on a precision balance. We performed three sets of experiments in three cylindrical containers with different sizes (diameters = 7, 4.5, and 3 cm) in order to vary the free surface area. The mass loss during the drop evolution was determined in each container for drops of increasing volume (3–26.5 µl) with 0.5 mmol l$^{-1}$ CTAB in each. Results obtained in the 7 cm container are plotted in Fig. 9. During these experiments, each drop exhibited pulsations or at least an unstable regime within which it emitted randomly smaller droplets from its leading edge. Starting as soon as the drop was deposited in the 7 cm container, the instability occurred after an induction period in smaller containers (except for the smallest drops). Nevertheless, the overall evolution of the mass loss is similar in all cases, exhibiting a quasi-linear decrease throughout the drop life time (the drop disappears at the instant of time indicated by an arrow in Fig. 9). The fact that the evaporation rate does not depend upon time is an indication that the main source of evaporation is not the drop surface itself (the area of which decreases in time) but essentially the DCM film surrounding it.

Plotting the evaporation rate (defined as the slope of the linear part of each curve in Fig. 9) as a function of the initial drop volume for the three different containers (Fig. 10a) reveals that this rate first exhibits a linear increase, then a constant value. The smaller the container, the lower the drop volume for which the plateau is reached. Hence, the maximum evaporation rate is determined by the free surface area, which becomes the size of the evaporating surface when the film that surrounds the drop (made of the macroscopic film over which the paper focuses but also of the precursor film) has invaded the entire container surface. Maximum evaporation rates plotted vs. the free surface area of the three containers (Fig. 10b) reveal a linear variation, providing the estimate of the evaporation flux, $Q_{ev} = 2 \times 10^{-4}$ mg mm$^{-2}$ s$^{-1}$. It may be noticed from Fig. 10a that the evaporation rate for a 5.6 µl drop in a 7 cm Petri dish is ~ 0.4 mg s$^{-1}$. With the above value of $Q_{ev}$, this implies that the free surface area involved in the evaporation process corresponds to a disk with a 25 mm radius. This is much larger than the 7–8 mm radius reached by the spreading film during each pulsation (see Fig. 2), underlining the major contribution of the precursor film in the overall evaporation rate.

**Derivation of the initial spreading law**. After the drop is released at the water surface, the spreading coefficient $S(t)$ has to relax toward $S_\infty$ for reasons discussed in the text. The relaxation process is dominated by desorption within the aqueous phase, so that the CTAB concentration at the water surface, $\Gamma(t)$, obeys the approximate evolution

$$\mathrm{d}\Gamma/\mathrm{d}t = k_a C(\Gamma_\infty - \Gamma) - k_d \Gamma, \qquad (2)$$

where $k_a$ and $k_d$ are the adsorption and desorption coefficients, respectively, $C$ is the CTAB concentration in the bulk, and $\Gamma_\infty$ is the surface concentration at critical micellar concentration (CMC). The solution of equation (2) is

$$\Gamma(t) = \Gamma_{eq} + (\Gamma(0) - \Gamma_{eq})\exp(-k_{eff}t), \qquad (3)$$

where $\Gamma_{eq} = \Gamma_\infty/(1 + k_d/k_a C)$ is the surface concentration at equilibrium and $k_{eff} = k_d + k_a C$ is the effective desorption coefficient. With $C$ much lower than the CMC, the surface tension at the oil–air interface varies linearly with $\Gamma$. Hence, the spreading coefficient undergoes the same exponential relaxation as $\Gamma$ toward its equilibrium value, i.e., $S(t) = S_\infty - \Delta S \exp(-k_{eff}t)$, which for times $t \ll k_{eff}^{-1}$, may be approximated as

$$S(t) \approx S_\infty - \Delta S + (k_{eff}\Delta S)t. \qquad (4)$$

Now, let $R_t = \mathrm{d}R/\mathrm{d}t$ be the instantaneous velocity of the leading edge of the film (throughout this section, $R(t)$ stands for the film radius $R_F(t)$ defined in Fig. 1a). For the sake of simplicity, we provisionally assume that the film thickness, $h$, is uniform and does not change over time, i.e., the drop 'feeds' the film in such a way that $\mathrm{d}h/\mathrm{d}t$ is negligibly small. Then the radial velocity $U$ within the film at a radial position $r$ reduces to $U(r,t) = R(t)R_t(t)/r$. The spreading is driven by the line force $2\pi R(t)S(t)$ acting at the leading edge. This force is counteracted by the viscous resistance of water, the depth of which is large compared with $h$. The corresponding shear stress at a radial position $r$ scales as $\mu_w U(r, t)/\delta(r, t)$, where $\delta$ is the local thickness of the boundary layer that grows over time below the film and $\mu_w$ is the water viscosity. The resisting force acting on the film is thus $2\pi\mu_w R(t)R_t(t)\int_0^{R(t)}\delta^{-1}(r, t)\mathrm{d}r$. Defining the average boundary layer thickness $\langle\delta\rangle(t)$ as $R(t)/\langle\delta\rangle(t) = \int_0^{R(t)}\delta^{-1}(r, t)\mathrm{d}r$, the total resisting force is found to scale as $2\pi\mu_w R^2(t)R_t(t)\langle\delta\rangle^{-1}(t)$, so that the force balance implies $\mathrm{d}(R^2)/\mathrm{d}t \sim \mu_w^{-1}S(t)\langle\delta\rangle(t)$. Since the average boundary layer thickness grows as $\langle\delta\rangle(t) \propto (\mu_w/\rho_w)^{1/2}$ ($\rho_w$ is the water density), one has

$$\mathrm{d}(R^2)/\mathrm{d}t \sim (\rho_w\mu_w)^{-1/2}S(t)t^{1/2} \qquad (5)$$

If the spreading coefficient is constant, equation (5) immediately yields the classical $R(t) \propto t^{3/4}$ law. Instead, if it varies linearly upon time, inserting (4) in (5) and integrating yields the 'super-spreading' prediction (1).

**Velocity and temperature fields within the spreading film**. We consider the drop as an infinite reservoir, localized at $r = 0$, which provides a constant flowrate $Q$ to the film and keeps a constant temperature, $T_D$. We now let the film thickness vary over time but, for simplicity, still assume that it is radially uniform. The mass of fluid contained within the film is then $\pi R^2(t)h(t)$ and we assume that its variations are only due to the incoming flowrate $Q$, i.e., $\mathrm{d}(\pi R^2 h)/\mathrm{d}t = Q$. In particular, we neglect the evaporation velocity, considering that it is much smaller than the rate of change of the film thickness, $h_t = \mathrm{d}h/\mathrm{d}t$. Assuming $R(t) \propto t^n$, the constant flow rate assumption implies $h(t) \propto t^{1-2n}$. Hence, $Rh_t/hR_t = (1 - 2n)/n$, so that $Q = \pi h R R_t/n$. The velocity field corresponding to the base flow within the film is then

$$\mathbf{u}(r, z, t) = \frac{RR_t}{2nr}\mathbf{e_r} + \frac{1 - 2n}{n}\frac{R_t}{R}\left(z\mathbf{e_z} - \frac{r}{2}\mathbf{e_r}\right) \qquad (6)$$

where $\mathbf{e_r}$ and $\mathbf{e_z}$ denote the unit vectors in the radial and vertical directions, respectively. Thus, the flow field is made of a source corresponding to DCM injection at the drop surface plus a linear strain resulting from the decrease of the film thickness.

In the energy balance, diffusion is negligible in the radial direction, as the aspect ratio $h/R$ is very small. For the same reason, convective transport by the vertical velocity $h_t z/h$ is much smaller than its radial counterpart. It will also be proved a posteriori that the time rate-of-change term is small. We may also neglect heat exchanges at the oil/water interface, considering that temperature gradients are essentially driven by evaporation. Last, following the conclusions of our measurements, we assume that DCM evaporates at a rate that is proportional to the area of the oil/air interface. The local evaporation flux, $J$, may vary with $r$ and $t$ but has a prescribed surface average, $J_0$ (corresponding to that determined experimentally), so that $2\pi\int_0^{R(t)}J(r, t)r\mathrm{d}r = \pi R^2(t)J_0$.

With these assumptions, the evolution of the temperature within the film is driven by the approximate heat equation $U(r,t)\partial_r T = \lambda_o\partial_{zz}T$, with boundary conditions $\partial_z T = 0$ at $z = 0$ and $-2\pi k_o\int_0^{R(t)}r\partial_z T(t, r, z = h)\mathrm{d}r = \pi R^2(t)J_0$, where

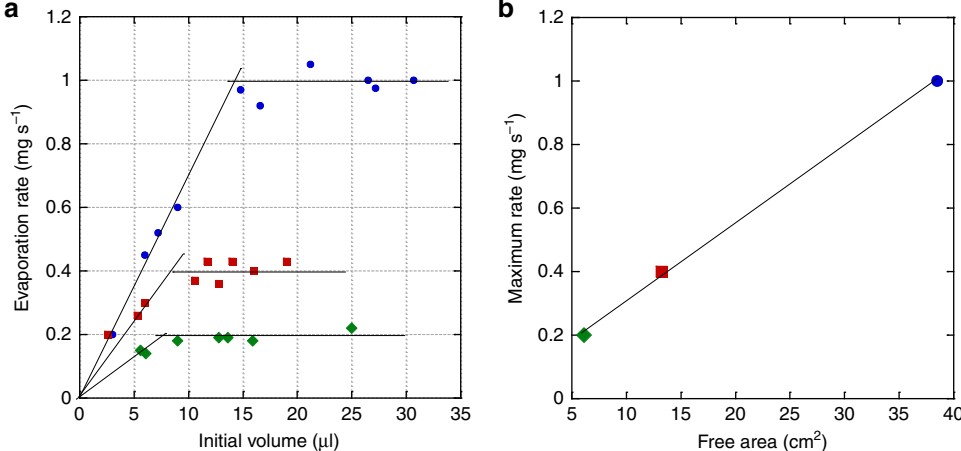

**Fig. 10** Evaporation rate in three containers with different diameters. **a** As a function of the initial drop volume; **b** maximum evaporation rate vs. free surface area. The containers diameters are 3 cm (green diamonds), 4.5 cm (red squares), and 7 cm (blue dots), respectively

$k_o$ and $\lambda_o = k_o/\rho_o C_{po}$ denote the oil thermal conductivity and diffusivity, respectively, $C_{po}$ being the heat capacity at constant pressure.

Provided that time variations of $T$ are small (see below) and neglecting the temperature variation due to the evaporation flux at the drop edge (which amounts to considering that $J_0 h^2/k_o RR_t \ll T_D$), the temperature distribution within the film is found to be

$$T(r,z,t) = T_D - \frac{2n}{2n+1}\frac{J_0}{k_o h RR_t}\left\{\lambda_o r^2 + RR_t\left(1 + \frac{2n-1}{2n}((r/R)^2 - 1)\right)z^2\right\} \tag{7}$$

The linear radial increase of $\partial_r T$ predicted by (7) is related to the circular symmetry of the present system, which is responsible for the $1/r$ variation of the source term in (6). With such a variation, supplying a uniform surface-averaged evaporation flux requires $\partial_r T$ to increase linearly from the edge of the drop to the film periphery. With $R(t) \propto t^n$ and $h(t) \propto t^{1-2n}$, it is readily seen that the neglected terms in the energy balance behave as $t^{-2n}$, whereas those that were kept behave as $t^{2n-1}$. Hence, the approximations leading to (7) are legitimate provided that $n > 1/4$, which is always satisfied in present experiments. Obviously, (7) breaks down when the constant flow rate condition $dQ/dt = d^2(R^2h)/dt^2 = 0$ is no longer satisfied, which happens when the transition from spreading to recoil takes place.

**Structure of the Marangoni flow.** Assuming that the oil/air surface tension $\gamma_{oil/air}$, now abbreviated as $\gamma_{oa}$, decreases linearly with temperature as $\gamma_{oa}(T) = \gamma_{oa}(T_D) - \alpha(T - T_D)$ $(\alpha > 0)$, (7) implies $\partial_r \gamma_{oa} = Kr$, with $K = 4\alpha n J_0\{(2n+1)\rho_o C_{po} h RR_t\}^{-1}$. The Marangoni stress associated with this surface tension gradient induces a correction to the base flow. Imposing that the vertical velocity component, $w_M$, of this correction vanishes at both $z = 0$ and $z = h$, and that the horizontal component, $u_M$, vanishes at $z = 0$ (which sounds reasonable since the viscosity of the water is ~2.5 larger than that of DCM), the velocity distribution in the Marangoni-induced flow obeys

$$u_M = \frac{K}{\mu_o}rz, \quad w_M = \frac{K}{\mu_o}(h-z)z \tag{8}$$

The horizontal component, $u_M$, remains weak up to the rim compared with the radial component of the primary velocity determined in (6) as far as the characteristic Marangoni velocity, $V_M = 2\{n\alpha J_0[(2n+1)\rho_o \mu_o C_{po}]^{-1}\}^{1/2}$, is much smaller than $R_t$.

**Rim structure and estimate of the film thickness.** In the spreading film, according to (6) and (8), the height-averaged velocity resulting from the combined effects of the spreading and the Marangoni effect is $RR_t/2nr + \{[(2n-1)/2n]R_t/R + Kh/2\mu_o\}r$. The film thickness $h(t)$ considered up to now does not vary with $r$. Hence, it may be viewed as the average of the actual film thickness over the entire area covered by the film. If for some reason the local film thickness, $h_{loc}(r,t)$, exhibits some variation with $r$, hydrostatic and capillary effects induce a radial pressure gradient, which in turn results in a correction to the above height-averaged velocity. The local pressure within the film is $p(z,$

$h_{loc}) = \rho_o g(h_{loc} - z) + \gamma_{oa}\kappa(h_{loc})$, $\kappa$ denoting the interface mean curvature, which, for a slowly varying $h_{loc}$, may be approximated as $\kappa \approx -\partial_{rr}h_{loc}$. The radial component, $u_c$, of the complementary flow induced by this local pressure gradient is then $u_c(r,z,t) = \mu_o^{-1}\,\partial_r\{\rho_o g h_{loc} - \gamma_{oa}\partial_{rr}h_{loc}\}z(z/2 - h_{loc})$, where it has been assumed that $u_c = 0$ at $z = 0$ and $\partial_z u_c = 0$ at $z = h$. This yields a depth-averaged velocity correction

$$<u_c> = \partial_r(\gamma_{oa}\partial_{rr}h_{loc} - \rho_o g h_{loc})h_{loc}^2/3\mu_o \tag{9}$$

This contribution is responsible for the existence of the rim, which is the narrow region within which the Marangoni flow adjusts to the flow ahead of the film, where a shear-free condition holds at the free surface[34]. For the matching to be possible within a rim with characteristic radius $a$, at least one of the contributions in (9) must be of the same order as the height-averaged Marangoni-induced velocity, $< u_M > \approx Krh_{loc}/2\mu_o$. This allows in principle for three possibilities, depending on which of the capillary ($\gamma_{oa}\partial_{rrr}h_{loc}$), Marangoni ($\partial_r\gamma_{oa}\partial_{rr}h_{loc}$), or gravity ($\rho_o g\partial_r h_{loc}$) term dominates. However, the dominant contribution is usually provided by the first of these, as it involves the derivative of highest order. Thus, balancing $< u_M >$ with the capillary term $\gamma_{oa}h_{loc}^2\partial_{rrr}h_{loc}/3\mu_o$ implies $a^3 \approx 2\gamma_{oa}h_{loc}^2/(3Kr)$, i.e., $(a/h_{loc})^3 \approx \gamma_{oa}/(3\mu_o<u_M>)$ which, on using (8), yields

$$\frac{h_{loc}}{a} \approx \left(\frac{6n}{2n+1}\frac{\alpha J_0}{\gamma_{oa}\rho_o C_{po}R_t}\right)^{1/3} \tag{10}$$

With this estimate at hand, it may be checked that the two neglected contributions in (9) are both < 1% of that of capillarity. As $R_t \propto t^{n-1}$ and $a \propto t^{-n/2}$, (10) implies that $h_{loc}$ varies as $t^{(2-5n)/6}$ close to the rim. This is significantly less than the $t^{1-2n}$ variation found for $h$ (for $n = 5/4$ and $3/4$, the exponents are $-17/24$ and $-7/24$, instead of $-3/2$ and $-1/2$, respectively). This difference is readily understood by considering that the local film thickness behaves as $h_{loc}(r,t) \propto r^p t^q$ in the vicinity of the rim. If so, the surface average of $h_{loc}$ is proportional to $R^p(t)t^q$, i.e., to $t^{pn+q}$, which implies $1 - 2n = pn + q$. Assuming $q = (2-5n)/6$ as predicted by (10) then yields $p = (4-7n)/6n$, i.e., $p = -19/30$ and $-5/18$ for $n = 5/4$ and $3/4$, respectively. This result shows that the film thins down as one moves radially outward, and this thinning is governed by evaporation, at least close to the rim.

Having determined experimentally the evaporation heat flux, $J_0 \approx 70$ W m$^{-2}$, and knowing the value of the physical parameters involved in (10) ($\alpha = 1.3 \times 10^{-4}$ N m$^{-1}$ K$^{-1}$, $C_{po} = 1.19 \times 10^3$ J kg$^{-1}$ K$^{-1}$, $\rho_o = 1.33 \times 10^3$ kg m$^{-3}$, and $\gamma_{oa} = 2.8 \times 10^{-2}$ N m$^{-1}$), the above prediction may be used to get an estimate of the film thickness close to the rim at a specific stage of the spreading. For instance, in Fig. 1a, the first stage where the rim is closest to the drop corresponds to $R \approx 3.75$ mm, $R_t \approx 50$ mm s$^{-1}$, and $a \approx 0.125$ mm. As $n \approx 1.1$ at this stage, (10) yields $h_{loc} \approx 2.7$ μm. Similarly, during $P_3$, the first stage displayed in Fig. 1b corresponds to $l \approx 45$ μm with $R \approx 5.5$ mm and $R_t \approx 30$ mm s$^{-1}$, which yields $h_{loc} \approx 1.1$ μm.

**The Rayleigh–Plateau mechanism of rim break-up.** The two snapshots displayed in Supplementary Fig. 2, shifted by a time interval $t_2 - t_1 \approx 14.9$ ms, both refer to a stage of pulsation $P_1$ where most of the circumferential variations displayed by the rim thickness are sufficiently small for the linear stability theory to apply

approximately (Supplementary Fig. 2a corresponds to the rim position closest to the drop in Fig. 1a). Considering well-defined varicosities with a moderate amplitude in these two images (see the caption of Supplementary Fig. 2 for more details on how we determined the corresponding wavelength $\Lambda$) yields $\Lambda(t_2)/\Lambda(t_1)$ = 1.245. Experimentally, we do not have access to the rim radius, $a$, defined such that the area of the rim cross section equals $\pi a^2(t)$ and the rim volume equals $2\pi^2 R(t)a^2(t)$. As the actual shape of the rim cross-section may not be exactly circular, owing to the oil-to-water density contrast and to capillary effects, it is important to check whether or not the rim thickness $2l$ determined from top views behaves similarly to the rim diameter $2a$. This may be assessed by examining the variation of the quantity $Rl^2$ between Supplementary Fig. 2a and b: the ratios $R(t_2)/R(t_1)$ and $l(t_2)/l(t_1)$ of the film radii and rim thicknesses are ~1.244 and 0.887, respectively, so that $R(t_2)l^2(t_2)/R(t_1)l^2(t_1) \approx 0.98$. Strict conservation of the rim volume implies $R(t_2)a^2(t_2)/R(t_1)a^2(t_1) = 1$. Hence, up to image accuracy, $l$ and $a$ behave similarly and it may safely be assumed that $l \approx a$, which allows us to conclude that $\Lambda/a$ is close to $\Lambda/l$ and has increased from 7.4 to 10.4 in between the two panels.

It must then be determined whether this increase stems from the intrinsic mechanisms that select the most unstable mode in the case of a stretched thread, or if this most unstable mode has already been selected by time $t = t_1$, in which case the above variation of $\Lambda/l$ from $t = t_1$ to $t = t_2$ merely results from the conservation of the number of wavelengths along the circular rim, due to its $2\pi$ periodicity. Assuming that the rim expands as $R(t) \propto t^n$, conservation of the number of wavelengths implies that the dominant wavelength evolves as $\Lambda(t) \propto t^n$. The above values of $R(t_2)/R(t_1)$ and $\Lambda(t_2)/\Lambda(t_1)$ are very close, confirming that $\Lambda(t) \propto R(t)$ during that stage of the rim evolution. Hence, the variation of $\Lambda(t)/l(t)$ observed between the two instants of time displayed in Supplementary Fig. 2 proves that the most unstable mode that eventually leads to rim break-up has already been selected by time $t = t_1$.

Although many images were taken at times $t < t_1$, we chose the snapshot corresponding to $t = t_1$, because it is the first in which disturbances with a well-defined wavelength clearly emerge along the entire rim perimeter. For this reason, we consider that the selection process from which the most amplified disturbance emerges is just completed by $t = t_1$. Hence, comparison with theoretical predictions must be conducted at that specific instant of time. To achieve this comparison, several additional parameters have to be known. First of all, it is necessary to determine the proper origin of time, which is the instant at which the rim volume becomes constant (in the earlier stages, the rim thickens by extracting fluid from the film, so that its volume increases). We plotted the evolution of the quantity $R(t)l^2(t)$ using images recorded for $t < t_1$ and determined that the rim volume reaches a plateau when $R_0 = R(t = t_0) \approx$ 3.45 mm. From that origin until $t = t_1$, the film radius evolves as $R(t) = R_0(t/t_0)^n$ with $n \approx 1.1$, so that the ratio $R(t_1)/R_0$ yields $(t_1 - t_0)/t_0 \approx 0.075$. Next, the characteristic times involved in the rim evolution have to be determined. The capillary and viscous time scales evaluated at $t = t_0$ are $T_\gamma = (\rho_o l^3(t_0)/\gamma_{oa})^{1/2} \approx 3.2 \times 10^{-4}$ s and $T_\mu = \rho_o l^2(t_0)/\mu_o \approx 5.35 \times 10^{-2}$ s, respectively. The rate at which the rim is stretched may be estimated by considering the variation of $\Lambda/l$ between two successive images and making use of the evolution law $\Lambda(t)/l(t) \propto (t/t_0)^{3n/2}$. With this procedure, the characteristic stretching time is found to be $T_s = t_0 \approx 4.95 \times 10^{-2}$ s. Hence at $t = t_0$, $T_\gamma/T_{\mu 0} \approx 6 \times 10^{-3}$ and $T_\gamma/T_s \approx 6.5 \times 10^{-3}$, both of which are $\ll 1$. This indicates that, at a given $t < t_1$, the instantaneous growth rate, $\omega(\lambda, t)$, of a disturbance with wavelength $\lambda(t)$ is essentially driven by capillary effects, with only a weak influence of stretching and viscous effects, which act over much longer time scales. In other terms, $\omega(\lambda,t)$ is the solution of the classical Rayleigh dispersion equation[35]. It is noteworthy that $\mu_w/\mu_o$ being ~2.4, the viscosity of the aqueous phase does not affect significantly $\omega(\lambda(t),t)$ either.

The theory closest to present experimental conditions was developed first in the inviscid limit[36] $T_{\mu 0}/T_s \to 0$, then in the presence of finite viscous effects[37]. This theory only considers the case where the stretching rate is constant, i.e., $n = 1$. However, as $n \approx 1.1$ throughout the $P_1$ pulsation, this restriction is not severe. In contrast, any influence of the surrounding medium is neglected in that theory, which does not fit well with present conditions where the rim is half-immersed in another more viscous liquid. For arbitrary values of $T_\gamma/T_s$ and $(t - t_0)/t_0$, the theory provides the value of the most amplified wavelength, $\Lambda(t)$, which is the one that maximizes the cumulated growth rate $\int_{t_0}^{t} \omega(\lambda(t'), t')dt'$. In the inviscid case, it predicts that in the limit $T_\gamma/T_s \to 0$, which is relevant here, the most amplified mode increases over time, starting from Rayleigh's classical prediction $\Lambda/l = 9.01$ at $t = t_0$ and reaching the value $\Lambda/l = 9.94$ at $t = t_1$. However, viscous effects due to the stretching process modify the normal viscous stress at the thread or rim surface and their influence is destabilizing at short time[37]. For this reason, at a given time such that $(t - t_0)/t_0 \lesssim 1$, these effects reduce the wavelength exhibiting the maximum cumulated growth rate. In particular, for $T_{\mu 0}/T_s = 1$, the initial $\Lambda/l$ is found to be reduced to 6.65. In present experiments, $T_{\mu 0}/T_s \approx 1.08$, which is close to unity and makes comparison with theoretical predictions obtained with $T_{\mu 0}/T_s = 1$ relevant. Supplementary Table 1 summarizes this comparison. It shows that the theory predicts $\Lambda/l = 7.53$ at $t = t_1$, which is remarkably close to the experimental observation $\Lambda/l = 7.4$. The 2% difference is probably fortuitous given the approximations involved in the experimental estimates and the extra influences not taken into account in the theory, especially the viscous effects due to the aqueous phase and the possible role of evaporation. However, the agreement is clearly more than qualitative and entirely supports the view that the mechanism driving the growth of circumferential disturbances along the rim is the Rayleigh–Plateau instability, which here acts on a continuously stretched rim in the presence of weak but non-negligible viscous effects.

**Evaporative instability in the circumferential direction.** Wrinkles start growing at the edge of the drop. Similar to what happens with the Rayleigh–Plateau instability at the rim surface, we consider that the $2\pi$ periodicity imposed by the circular geometry has no significant influence on their growth. In contrast, once the most amplified wavelength is selected, its variation with the radial position simply stems from the fact that the number of wrinkles must stay constant whatever $r$. For this reason, the following stability analysis is carried out at $r = 0$, where we examine the possibility for periodic disturbances to develop at the film surface in the spanwise direction, $y$, representing the circumferential direction in the real system. We consider a disturbance with wavenumber $k_y$ and growth rate $s$, and assume that the free surface at $r = 0$ takes the form $h_0(y,t) = h_1 + \eta(y,t)$, with $h_1 = h_{loc}(r = 0,t)$ and $\eta(y,t) = \eta_0 \exp(ik_y y)e^{st}$ ($i^2 = -1$). We adopt the long-wave approximation, $k_y h_0 \ll 1$, and restrict the analysis to the small-amplitude limit, $\eta_0/h_0 \ll 1$. Similar to the reasoning that led to (9), hydrostatic, capillary, and Marangoni effects induced by the disturbance result in pressure and surface tension gradients in the $y$ direction. Thus, the profile of the disturbance-induced spanwise velocity component, $v_\eta$, is

$$v_\eta(y,z,t) = \mu_o^{-1}\{\partial_y(\rho_o g\eta - \gamma_{oa}\partial_{yy}\eta)z(z/2 - h_0) + \tau(\eta)z\} \quad (11)$$

From continuity, the vertical velocity component, $w_\eta$, associated with $v_\eta$ is found to be

$$w_\eta(y,z,t) = -\mu_o^{-1}\{\partial_{yy}(\rho_o g\eta - \gamma_{oa}\partial_{yy}\eta)z^2(z/3 - h_0)/2 + \partial_y\tau(\eta)z^2/2\} \quad (12)$$

The disturbance $(v_\eta, w_\eta)$ takes the form of a periodic cellular flow in the $(y,z)$ vertical plane. The shear stress at the free surface, $\tau(\eta) = \partial_y\gamma_{oa}$, results from variations of the surface temperature $\partial_y T(r = 0,z = h_0(y,t),t)$ due to spanwise variations $\partial_y\eta$ of the film thickness. From (7) taken at $r = 0$, we find

$$\tau(\eta) = \frac{2}{2n+1}\alpha\frac{J_0}{k_o}\partial_y\eta \quad (13)$$

Injecting (13) into (11) and (12) indicates that, provided the last term in the latter two equations dominates, $w_\eta$ is positive when $\eta$ is maximum, i.e., an upwelling motion takes place below the crests of the film and the circumferential component $v_\eta$ brings fluid away from them at the free surface. Mass conservation across the film requires that $\partial_t\eta + \partial_y\int_0^{h_0(y,t)}v_\eta(y,z,t)dz = 0$. Making use of (11) and (13) into this mass balance yields the evolution equation for the film height

$$\partial_t\eta + \partial_y\left\{\partial_y(\gamma_{oa}\partial_{yy}\eta - \rho_o g\eta)\frac{h_0^3}{3\mu_o} + \frac{1}{2n+1}\alpha\frac{J_0 h_0^2}{\mu_o k_o}\partial_y\eta\right\} = 0 \quad (14)$$

The existence of possible non-trivial solutions is assessed by considering the linearized form of (14), namely

$$\partial_t\eta + \partial_y\left\{(\gamma_{oa}\partial_{yyy}\eta - \rho_o g\partial_y\eta)\frac{h_1^3}{3\mu_o} + \frac{1}{2n+1}\alpha\frac{J_0 h_1^2}{\mu_o k_o}\partial_y\eta\right\} = 0 \quad (15)$$

which may be recast in the generic form

$$\partial_t\eta + A\partial_{yy}\eta + B\partial_{yyyy}\eta = 0 \quad (16)$$

$$\text{with} \quad A = \frac{1}{2n+1}\alpha\frac{J_0 h_1^2}{\mu_o k_o} - \rho_o g\frac{h_1^3}{3\mu_o}, B = \gamma_{oa}\frac{h_1^3}{3\mu_o} \quad (17)$$

Setting $\eta(y,t) = \eta_0\exp(ik_y y)e^{st}$, (16) predicts the growth rate of the disturbance to be

$$s = Ak_y^2 - Bk_y^4 \quad (18)$$

As $B > 0$, disturbances may grow only if $A > 0$, in which case wavenumbers in the range $0 \le k_y \le (A/B)^{1/2}$ are amplified. For $A$ to be positive at the edge of the drop, one needs $\alpha J_0/k_o > (2n+1)\rho_o gh_1/3$, i.e., the local undisturbed film thickness $h_1$ has to be less than a critical value, $h_c = 3\alpha J_0[(2n+1)\rho_o gk_o]^{-1}$. With $k_o = 0.14$ W m$^{-1}$ K$^{-1}$ and $J_0 \approx 70$ W m$^{-2}$, this yields $h_c \approx 4.3$ μm and $h_c \approx 6$ μm for $n = 5/4$ and $3/4$, respectively.

In films thinner than this critical value, (18) implies that the most amplified wavenumber is $k_{max} = (A/2B)^{1/2}$, so that at the drop edge, the number of wrinkles, $N_W = k_{max}R_D$, that emerge should be close to

$$W \approx R_D\left(\frac{\rho_0 g}{2\gamma_{oa}}\right)^{1/2}\left(\frac{h_c}{h_1} - 1\right)^{1/2}, \quad (19)$$

i.e., $W \approx 1.3\left(\frac{h_c}{h_1} - 1\right)^{1/2}$ with $R_D \approx 2.7$ mm. Using (10), we previously obtained an estimate of $h_{loc}$ valid close to $r = R$. Assuming that the self-similar evolution $h_{loc}(r,t) \propto r^p t^q$ applies throughout the entire range of radial positions with the previously determined exponents $p$ and $q$, we may estimate $h_1$ as $h_1(t) = (R_D/R(t))^p h_{loc}(R(t))$. Making use of this procedure at the stage corresponding to the rim with smallest

radius in Fig. 1a, b, one finds $h_l \approx 3.4$ µm and 1.7 µm for pulsations $P_1$ and $P_3$, respectively. Both values are lower than $h_c$ and (19) predicts $W \approx 0.9$ and $W \approx 1.6$, respectively. Turning to $P_4$, by the time wrinkles are first detected, the rim thickness is about 20 µm, so that relying on (10) and on the above procedure linking $h_l$ and $h_{loc}(R)$, one finds $h_l \approx 0.2$ µm, for which (19) predicts $W \approx 7$.

**Estimate of the inner rim lifetime.** We argue that the second rim that forms just before the recoil starts is not directly observed, owing to its very short lifetime. To prove this, it must first be reminded that nonlinear stability theory predicts[45] and experiments confirm[46] that an inviscid cylindrical jet with radius $a$ submitted to a 1% amplitude initial disturbance (which sounds a reasonable magnitude here) with wavelength $\lambda$ corresponding to the most amplified linear mode (i.e., $\lambda/a \approx 9.01$) breaks up within a time $T_{BU} \approx 15\, T_\gamma$, where $T_\gamma$ is the capillary time (here $T_\gamma = (\rho_o a^3/\gamma_{oa})^{1/2}$). Based on the size of the droplets that form the inner ring and considering that their volume equals $\pi \lambda a^2$, an estimate of $a$ may be obtained by assuming $\lambda/a \approx 9$, in agreement with Rayleigh's prediction[35]. These drops having diameters of ~ 100 and 35 µm, this reasoning suggests that $a$ is about 25 and 10 µm for pulsations $P_1$ and $P_3$, respectively. Thus, the rim break-up time for these two pulsations may respectively be estimated to 0.4 and 0.1 ms.

**Data availability.** All data are available from the corresponding author upon reasonable request.

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

## Acknowledgements

This work was supported by the DROPDYN project from the 2014 Programme Transversalité of the IDEX UNITI from Université Fédérale Toulouse Midi-Pyrénées (grant CNRS 127208). V.P. acknowledges the financial support from the Centre National d'Etudes Spatiales. We thank Sébastien Cazin for his expert technical assistance with the optical and high-speed imaging acquisition systems.

## Author contributions

F.W. and J.S. designed the experimental setup. F.W. carried out the experiments. V.P. directed the project, processed experimental images and supervised data analysis. J.M.

developed the theoretical analysis. V.P. and J.M. wrote the manuscript. All authors contributed to interpret experimental observations.

## Additional information

**Competing interests:** The authors declare no competing financial interests.

