## [Peer Review File · Nature Communications]

Reviewers' comments:

Reviewer #1 (Remarks to the Author):

This is an excellent paper. Anyone reading it must appreciate the beauty of nature and how complex but ordered behavior can arise from simple systems. The paper does an excellent job describing the phenomena and introduces the processes that very likely underlie the behavior of the system. A complete theory explaining the intricate behavior is beyond the scope of this paper.

Note the behavior here is more complex than other Marangoni based spreading I know of and it adds to that literature.

I recommend immediate publication.

Reviewer #2 (Remarks to the Author):

"When Marangoni effects draw a flower" by Wodlei et al.

This short paper reports the spreading behaviour of a drop of dichloromethane deposited on the surface of an aqueous phase in the presence of CTAB. The surface tension gradients generated by the drop deposition induce Marangoni instabilities that produce a spectacular pulsating regime. This is a novel and interesting observation, clearly described and supported by spectacular videos. The experiments are correctly interpreted, and the explanations provided are plausible.

The main weakness of this work is that it remains essentially qualitative without any attempt to quantitatively describe the collective behavior pattern using an appropriate theoretical framework. Numerous examples of Marangoni stresses that result in film spreading, dewetting, and fingering have been reported for which theories have been developed (in papers cited by the authors) that can certainly be applied to this particular case. Even if the observed phenomenon is difficult to comprehend in its globality, partly because of the complexity of the system with its different species that likely exchange between aqueous and drop phases, an effort could be done to separate its successive temporal sequences in order to submit them to theoretical treatment.

Furthermore, the literature presentation is too brief and somewhat vague. The paper would benefit from having a more straight-to-the-point introduction and a clear statement of the novelty of the findings.

In summary, this paper describes very interesting observations, but is too preliminary to be published in its present form. The observations are novel enough to warrant publication in *Nat. Commun.* if some theoretical treatment is provided and the novelty of the observations firmly established.

Response to Referee 1

This is an excellent paper. Anyone reading it must appreciate the beauty of nature and how complex but ordered behavior can arise from simple systems. The paper does an excellent job describing the phenomena and introduces the processes that very likely underlie the behavior of the system. A complete theory explaining the intricate behavior is beyond the scope of this paper. Note the behavior here is more complex than other Marangoni based spreading I know of and it adds to that literature.

I recommend immediate publication.

We were of course extremely pleased to hear the enthusiastic appreciation of the Referee and greatly thank him/her for his/her supportive recommendation.

The other referee was also mostly supportive but recommended that we undertake theoretical developments aimed at rationalizing at least some steps of our observations. This is why the new version is substantially longer than the original manuscript.

Regarding experiments, we think it is worth mentioning that we now describe (in the Methods section) a specific series of measurements by which we determined the evaporation rate which, as we now show, is the key player in the emergence of the observed pattern.

Regarding the theoretical analysis (developed in the Methods section), we focused on five central aspects of the physical processes governing the behaviour of the system, which now makes us in position to:

- ✓ explain the “super-spreading” regime observed during the expansion stage of each pulsation (grounded on detailed measurements synthesized in the new figure 7),
- ✓ (2) determine the dominant contributions to the velocity and temperature fields within the expanding film,
- ✓ (3) elucidate the connection between the rim thickness, the evaporation flux and the film thickness, by which we now obtain an indirect estimate of the latter,
- ✓ (4) demonstrate that rim break-up takes place through a Rayleigh-Plateau instability in the presence of weak but non-negligible stretching and viscous effects,
- ✓ (5) predict, through a linear stability analysis, that wrinkles are due to an evaporative instability and that they can only occur when the film is thinner than a critical value which we find to agree well with our observations.

The second referee also suggested that the introduction should be made more “straight-to-the-point”, which we achieved by (i) re-writing entirely the abstract, where we now list the main mechanisms yielding the observed pattern, and (ii) focusing the introduction, so as to position more directly the distinctive features of the present system in the context of the existing literature.

We hope that the Referee will enjoy these changes, especially the dissection of the fundamental mechanisms carried out in the “Physical mechanisms at play” section. We believe that the successive and inter-related mechanisms discussed there in the light of our theoretical developments provide a firm basis to set up a rational scenario explaining how the observed spectacular pattern emerges.

Response to Referee 2

This short paper reports the spreading behaviour of a drop of dichloromethane deposited on the surface of an aqueous phase in the presence of CTAB. The surface tension gradients generated by the drop deposition induce Marangoni instabilities that produce a spectacular pulsating regime. This is a novel and interesting observation, clearly described and supported by spectacular videos. The experiments are correctly interpreted, and the explanations provided are plausible.

We thank the Referee for his/her interest in our observations and positive appreciation on our analysis.

The main weakness of this work is that it remains essentially qualitative without any attempt to quantitatively describe the collective behavior pattern using an appropriate theoretical framework. Numerous examples of Marangoni stresses that result in film spreading, dewetting, and fingering have been reported for which theories have been developed (in papers cited by the authors) that can certainly be applied to this particular case. Even if the observed phenomenon is difficult to comprehend in its globality, partly because of the complexity of the system with its different species that likely exchange between aqueous and drop phases, an effort could be done to separate its successive temporal sequences in order to submit them to theoretical treatment.

We sincerely thank the Referee for the stimulation he/she provided with the above comment. Following this comment, we carried out a major revision of the paper aimed at providing a rational scenario explaining how the observed pattern emerges, based on theoretical developments for its most crucial steps.

Before we summarize these developments, we wish to point out to the Referee's attention that we also carried out a new series of specific experiments that allowed us to estimate the evaporation flux which, as we now show, is the key player in the emergence of the observed pattern.

The theoretical analysis now included in the paper (mostly in the "Methods" section, with direct implications to the experimental system discussed in the "Physical mechanisms at play" section) addresses five central aspects of the physical processes that contribute to establish its unique behaviour. Thanks to these theoretical developments, we now:

- ✓ (1) explain the "super-spreading" regime observed during the expansion stage of each pulsation (grounded on detailed measurements of the instantaneous film radius synthesized in the new figure 7),
- ✓ (2) provide the spatial distribution of the dominant contributions to the velocity and temperature fields within the expanding film,
- ✓ (3) elucidate the connection between the rim thickness, the evaporation flux and the film thickness, by which we obtain an indirect estimate of the latter,
- ✓ (4) demonstrate that rim break-up takes place through a Rayleigh-Plateau instability in the presence of weak but non-negligible stretching and viscous effects,
- ✓ (5) predict, through a linear stability analysis adapted from Hosoi & Bush (2001), that wrinkles are due to an evaporative instability and that they can only occur when the film is thinner than a critical value which we find to agree well with our observations.

Thanks to these additions, and to rational arguments explaining (i) why rims (with much smaller thicknesses than that of the primary expanding rim) appear repeatedly during film recoil and (ii) why they break up within a very short time, yielding the formation of rings and lines of droplets, we are now in position to propose a complete and rational scenario explaining how the observed pattern emerges.

Furthermore, the literature presentation is too brief and somewhat vague. The paper would benefit from having a more straight-to-the-point introduction and a clear statement of the novelty of the findings.

We agree that the abstract and the introduction could be made more “straight-to-the-point”. To this end we (i) entirely re-wrote the abstract, in which we now list the main mechanisms yielding the observed pattern; (ii) deeply re-focused the introduction, so as to position more directly the distinctive features of the present system in the context of the existing literature. We achieved these changes by successively considering the two major characteristics of our setup. First, having pointed out that DCM evaporation is the driving mechanism responsible for the Marangoni flow and the Marangoni instability that gives rise to the wrinkles, we briefly review some of the most closely related literature (other relevant works, such as those of Dussaud & Troian (1998) or Vuillemier *et al.* (1995) are later quoted in due course). Then, having mentioned that the present system exhibits unusually large and regular pulsations, we discuss similar systems and conditions required for the re-initialization of the spreading process to take place.

We believe that, with these changes, readers will more quickly be in position to figure out what the specificities of the physical and chemical processes later described in the paper are, and what the novelty of our system compared to those described in the available literature is.

In summary, this paper describes very interesting observations, but is too preliminary to be published in its present form. The observations are novel enough to warrant publication in Nat. Commun. if some theoretical treatment is provided and the novelty of the observations firmly established.

Again, we thank the Referee for having considered that, provided the paper were suitably strengthened, it would deserve to be published in *Nature Communications*. We hope that, with the major theoretical additions provided in the revised version and the careful re-writing of the abstract and introductory section, he/she will find that this version now makes a significant addition to the field and is suitable for publication.

REVIEWERS' COMMENTS:

Reviewer #2 (Remarks to the Author):

After reading the letter from the authors explaining the changes done to take into account my recommendations, as well as the revised manuscript, I consider that this work can now be published in Nat. Communications. The extra work has significantly benefited to the paper, which is now very satisfactory.

Congratulations to the authors!